# Functional Genetics to Understand the Etiology of Autoimmunity

**DOI:** 10.3390/genes14030572

**Published:** 2023-02-24

**Authors:** Hiroaki Hatano, Kazuyoshi Ishigaki

**Affiliations:** Laboratory for Human Immunogenetics, RIKEN Center for Integrative Medical Sciences, Yokohama 230-0045, Japan

**Keywords:** V2F, immunogenetics, TCR, eQTL, sQTL

## Abstract

Common variants strongly influence the risk of human autoimmunity. Two categories of variants contribute substantially to the risk: (i) coding variants of *HLA* genes and (ii) non-coding variants at the non-*HLA* loci. We recently developed a novel analytic pipeline of T cell receptor (TCR) repertoire to understand how *HLA* coding variants influence the risk. We identified that the risk variants increase the frequency of auto-reactive T cells. In addition, to understand how non-coding variants contribute to the risk, the researchers conducted integrative analyses using expression quantitative trait loci (eQTL) and splicing quantitative trait loci (sQTL) and demonstrated that the risk non-coding variants dysregulate specific genes’ expression and splicing. These studies provided novel insight into the immunological consequences of two major genetic risks, and we will introduce these research achievements in detail in this review.

## 1. Introduction

The genome-wide association study (GWAS) aims to detect associations between germline genetic variants and human phenotypes. The GWAS has no reverse causation: the phenotype cannot affect the variant. Therefore, the GWAS is one of a few studies that can assess the causal mechanism of human diseases. Over the past ten years, large-scale GWASs for autoimmune diseases have successfully detected hundreds of risk variants, exemplified by studies for rheumatoid arthritis (RA) [1,2] and systemic lupus erythematosus (SLE) [3,4,5]. However, the primary GWAS outputs are just a group of statistics of genome-wide variants. To extract biological information from GWAS results, we first need to extensively conduct genetic studies that connect variants to function (V2F). We then can infer the causal mechanisms of human autoimmunity by integrating GWAS and V2F study results (Figure 1). In this review, we provide various V2F studies and show how such study contributed to a better understanding of human autoimmunity etiologies.

### 1.1. Genetic Risk by HLA Coding Variants

The most outstanding characteristic of the GWAS for autoimmune diseases is the striking associations at the major histocompatibility complex (MHC) region, reflecting coding variants of *HLA* genes (Figure 2). Previous studies reported the *HLA* genes’ risk and protective amino acid polymorphisms [6,7,8,9,10]. For example, the risk of RA is strongly associated with *HLA-DRB1**0401 in European ancestries and *HLA-DRB1**0405 in East Asian ancestries; and *HLA-DRB1**1501 has been associated with multiple sclerosis (MS). Using sophisticated analytical strategies, researchers fine-mapped the MHC associations and demonstrated that a few amino acid positions of *HLA* genes account for most associations at the MHC locus. Raychaudhuri et al. reported amino acid polymorphisms at position 13 (or position 11 in strong linkage disequilibrium (LD) with position 13), 71, 74 of *HLA-DRB1*, position 9 of *HLA-B*, and position 9 of *HLA-DPB1,* which almost completely explain the MHC association to RA [11]. Intriguingly, all positions are located in peptide-binding grooves of *HLA* genes. Hu et al. conducted a similar analysis, and the top hit was found at position 57 of *HLA-DQB1*, followed by positions 13 and 71 of *HLA-DRB1* [12].

The canonical function of *HLA* genes is to present antigenic peptides to T cell receptors (TCR). Since the genetic risks are accumulated at the amino acid positions within peptide-binding grooves of *HLA* genes, we need to design V2F studies in the context of three players: *HLA*, antigenic peptides, and TCR. The etiological importance of this context is also supported by the fact that risk variants have been identified around genes encoding TCR signaling molecules. A good example is a missense variant of *PTPN22* (rs2476601; W620R), which is the top hit in RA-GWAS outside of the MHC region and shows pleiotropic associations for multiple autoimmune diseases [1,3,14]. *PTPN22* plays a key role in TCR signaling and inhibits T cell activation by dephosphorylating substrates involved in TCR signaling. Additionally, the genes implicated in RA GWAS were enriched for the TCR signaling pathway [15].

TCR is an “eye” of T cells, distinguishing self and foreign antigens. TCR can recognize antigenic peptides only when the peptides are presented on the HLA molecules. T cell initiates antigen-specific immune reactions involving multiple immune cell populations. Dysregulation of antigen-specific immunity is a hallmark of autoimmune diseases because we observe specific autoantibodies in the serum of autoimmune disease patients, e.g., anti-citrullinated peptide antibodies (ACPA) for RA and anti-double strand DNA antibodies for SLE [16,17]. In addition to the genetic evidence, this immunological evidence also supports the critical roles of *HLA*, antigenic peptides, and TCR in the pathology of autoimmunity. Since TCR signaling is a crucial factor for T cell development, activation, and differentiation, the antigenic peptide-HLA complex continuously influences T cells throughout the entire life cycle of T cells [18]. Therefore, V2F studies need to aim at different T cell developmental phases.

Historically, researchers have been conducting V2F studies mainly focusing on *HLA* and antigenic peptides by testing each *HLA* allele’s binding affinity to the pathogenic epitopes. The antigen-binding groove of the *HLA* class II molecule possesses several binding pockets accommodating the side chains of the antigenic peptides; the pockets with strong interaction are P1, P4, P6, P7, and P9 [19]. The idea is that when the pathogenic epitopes are more frequently presented to T cells in the peripheral tissues (e.g., inflammation sites and regional lymph nodes), the risk of developing autoimmunity should increase, which was introduced as the “peripheral hypothesis” of the *HLA* genetic risk in our recent article [13] (Figure 2).

In RA for instance, the high binding affinity of citrullinated epitopes, the most established pathogenic epitopes in RA, is found for the *HLA-DRB1* proteins encoded by the risk alleles. Scally et al. reported the structural basis of how the risk *HLA-DB1* alleles enhance an autoimmune reaction to the citrullinated epitopes, focusing on two *HLA* risk alleles (*HLA-DRB1*04:01* and *04:04*) with an electropositive P4 antigen-binding pocket and a protective allele (*HLA-DRB1**04:02) with an electronegative P4 pocket [20]. They demonstrated that *HLA-DRB1**04:01/04 with the positive P4 pocket favors citrulline (no net charge) but disfavors arginine (positively charged), whereas *HLA-DRB1**04:01/04 with the negative P4 pocket favors arginine. They also provided in-depth mass spectrometry analyses of the peptide repertoire bound to each *HLA-DR* allele and identified substantially different binding motifs, especially at the P4 pocket, where arginine was depleted in the risk alleles while tolerated in the protective allele. Hill et al. reported *HLA-DRB1**0401 transgenic mice immunized with cartilage proteoglycan aggrecan epitopes with arginine at P4 and those with citrulline at P4 [21]. They demonstrated that the arginine to citrulline conversion at P4 significantly increases peptide-HLA affinity and leads to activating CD4^+^ T cells in their transgenic mice.

Similarly, other studies also suggested the importance of the high binding affinity of the HLA risk alleles to the pathogenic epitopes in other autoimmune diseases such as type 1 diabetes (T1D) [22] and celiac disease [23].

Notably, the previous studies investigating the “peripheral hypothesis” did not consider how TCR repertoire is constructed before T cells encounter the molecular complex of *HLA* and pathogenic epitopes. T cells differentiate and mature in the thymus, where TCR is generated by random recombination. Thymic immature T cells randomly select and combine one TCR component gene from multiple candidates for each of V, D (only for β chain), and J gene while randomly adding or deleting several nucleotides at the junctional region of these component genes. This junctional region is called complementarity determining region 3 (CDR3). Due to these random processes, each T cell has a unique CDR3 sequence, which is a “fingerprint” of the T cell, and each human has a strikingly diverse repertoire of CDR3. Since CDR3 directly contacts with antigenic epitopes presented on the HLA molecule, the various CDR3 sequence patterns enable the immune system to recognize a wide range of antigens.

Reasonably, these random processes generate many non-functional TCRs that cannot interact with self-HLA molecules. Since TCR is an essential molecule for T cells, the thymus needs to select cells with functional TCR, called positive selection. Naturally, many of the T cells selected in this way are autoreactive, at least to some extent. To prevent autoimmunity, the thymus must eliminate T cells with TCR showing strong reactivity to autoantigens, called negative selection. These thymic selections drastically alter TCR repertoire, and most importantly, the peptide-HLA molecular complex has a critical role in these processes. Therefore, HLA risk alleles may affect the thymic selection and modify the TCR repertoire enhancing the autoreactivity, which is the “central hypothesis” of the HLA genetic risk [13] (Figure 2).

Motivated by this idea, we recently conducted the first genetic study testing associations between *HLA* alleles and TCR-CDR3 amino acid compositions, named cdr3-QTL [13]. In our research question, the TCR-CDR3 data (the response variable) are sequence data, and the *HLA* genotypes (the explanatory variable) are multi-allelic. Hence, the classical linear models were not feasible in this study. Therefore, we developed a novel analytical pipeline. First, we transformed CDR3 sequence data into a group of quantitative traits: a 20-dimensional vector with each component representing the usage frequency of each amino acid at a specific CDR3 position. We next transformed multi-allelic *HLA* genotype data (for example, m alleles) into a multi-dimensional vector, with each component representing the count of each *HLA* allele at a specific HLA position. We then applied a multivariate multiple linear regression model (MMLM) to detect associations between the CDR3 and *HLA* vectors, assessing the significance with the multivariate analysis of variance (MANOVA) test. Intuitively, this MMLM model estimates the correlation between CDR3 amino acid composition at a CDR3 position and all *HLA* alleles at an HLA position.

We applied our cdr3-QTL pipeline to publicly available TCR repertoire data of whole T cells from 628 healthy donors [24]. We demonstrated the strongest association at the amino acid position 13 of *HLA-DRB1*, the position with the strongest associations for the RA risk, and the 2nd strongest associations for T1D risk. These cdr3-QTL signals were successfully replicated in naïve CD4^+^ T cell TCR repertoire (number of donors = 169), and the signals were attenuated when we included clonally expanded T cell fraction. Therefore, the cdr3-QTL signals probably reflect thymic T cell selection rather than T cell selection during peripheral memory formation. Since the exact *HLA* position showed the most robust associations both for autoimmunity and CDR3 amino acid compositions, the *HLA* genetic risk is probably mediated by the thymic TCR-CDR3 selection dysregulated by *HLA* risk alleles.

In addition, we further conducted in-depth analyses to identify specific CDR3 patterns associated with *HLA* risks. We found several disease-specific patterns. RA and T1D *HLA* risk alleles increase acidic amino acid and decrease basic amino acid at the center of CDR3, linking the CDR3 negative charge and the genetic risk. In contrast, celiac disease *HLA* risk alleles increase hydrophobic amino acid at the center of CDR3. Previous studies showed that both amino acid charge and hydrophobicity of CDR3 influence antigen specificity [25]. Therefore, we hypothesized that accumulating these CDR3 amino acid patterns increases the T cell reactivity to pathogenic epitopes. We confirmed the possibility of this hypothesis by analyzing TCR sequence datasets derived from T cell subsets showing reactivity to several pathogenic epitopes: gluten-specific TCRs from celiac disease patients and citrullinated peptide-specific TCRs from RA patients. In summary, our study demonstrated striking associations between the *HLA* alleles and TCR-CDR3 amino acid compositions, providing novel genetic evidence supporting the “central hypothesis”.

### 1.2. Genetic Risk by Non-HLA Non-Coding Variants

In contrast to the *HLA* genes with a limited number of high-impact risk variants, non-*HLA* genes have numerous low-impact risk variants [26]. Specifically, the risk variants of non-*HLA* genes are enriched in the regulatory regions of relevant immune cell subsets. For example, the RA risk variants are enriched in the active regulatory regions of CD4^+^ T cell lineages, such as regulatory T cells [27,28]. Therefore, researchers have been conducting V2F studies to elucidate how variants affect the gene regulatory machinery in a cell type-specific manner (Figure 2).

The most straightforward scenario of the risk variant etiology is that they affect gene expression, i.e., expression of quantitative trait loci (eQTL). Therefore, researchers have conducted large-scale eQTL studies of immune cell subsets trying to illuminate the risk variant’s mechanisms, e.g., for which gene(s) and in which cell subset(s) the risk variants exert gene regulatory functions. The first wave of such research effort includes The Immune Variation (ImmVar) project, aiming to map the extent of variation in immune function in healthy human subjects [29]. Among multiple accompanying studies, Raj et al. conducted an eQTL study using purified CD4^+^ T cells and monocytes of 461 healthy donors, linking RA risk variants with T cell-specific eQTLs and Alzheimer’s disease risk variants with monocyte-specific eQTLs [30].

As the eQTL study platform matured, researchers started aiming to obtain a landscape of immune cell-specific eQTL across various immune cell subsets. We conducted an eQTL study using six immune cell subsets from 105 healthy donors [31]. Schmiedel et al. used 13 immune cell subsets isolated from 106 healthy donors [32] (DICE project). These research efforts were followed by our latest study that used 28 distinct immune cell subsets from 416 donors [33] (ImmuNexUT project). This study found several cell type-specific eQTLs colocalized with risk variants. For example, we observed the eQTL effect on ARHGAP31 only in plasmablasts, and the eQTL signal showed strong colocalization with a GWAS signal of SLE.

The intriguing characteristic of the ImmuNexUT project is that 337 among 416 donors were patients diagnosed with ten categories of immune-mediated diseases (IMD). Therefore, we were able to investigate the context-dependent eQTLs, e.g., how immune alterations in IMD patients affect eQTL effect size magnitude. We searched for genes whose expression level interacts with the eQTL effect; we called such genes ‘‘proxy genes’’ (pGenes). We successfully identified 37,875 significant pGene-eQTL interactions (FDR < 0.05). Furthermore, we found that pGenes were significantly overlapped with IFN signature genes, suggesting IFN has a pivotal role in the gene regulatory machinery in IMD patients. In addition, we found the enrichment of context-dependent eQTLs in GWAS top signals compared with all immune cell eQTLs.

Since the cell type specificity of eQTL signals is the key factor to elucidate the genetic etiology of complex traits, one of the most promising directions of eQTL research is arguably the single-cell level analysis as in other research fields. Monique et al. reported the first single-cell eQTL study using peripheral blood mononuclear cells (PBMC). Although the study scale is relatively limited (~25,000 PBMCs from 45 donors), they successfully showed the feasibility of a single-cell eQTL study, which produces very sparse expression data with many dropouts. They used the “pseudo-bulk approach” to mitigate this issue. They first conducted clustering to identify cell groups with similar expression profiles and created one expression data by integrating all cells within the same group (the data structure at this stage is essentially identical to that of bulk eQTL studies) and finally tested associating between genotypes and pseudo-bulk expression data. The pseudo-bulk approach is efficient and flexible. For example, this approach enables us to deploy previously established analytical pipelines for bulk eQTL, e.g., normalization, association tests, and the detection of cell type specificity of eQTL signals.

Using the pseudo-bulk approach, Perez et al. conducted a large-scale single-cell eQTL study using around 1.2 million PBMCs from 162 SLE cases and 99 healthy controls [34]. Among 3331 genes with at least one cis-eQTL in a cell type (FDR < 0.05), they identified 535 genes with at least one cell type-specific cis-eQTL. In addition, they reported several examples of colocalizations between single-cell eQTL and SLE-GWAS signals. One example is *ORMDL3,* a regulator of sphingolipid biosynthesis and ubiquitously expressed across cell types but showed eQTL-GWAS colocalization specifically in B cells, CD8^+^ T cells, and plasmacytoid dendritic cells with sufficiently high posterior probabilities (>90%).

Since the single-cell eQTL study is a relatively new field, its analytical strategy has not yet been fully matured. A promising alternative approach is the association test preserving single-cell resolution data structure. Three major hurdles for this approach are the sparsity in the expression data, the multiple repeated measurements from a donor, and a substantial amount of experimental noise. Assuming the expression count data follows a Poisson distribution, we can mitigate all hurdles using a Poisson mixed effects (PME) model with appropriate covariates to adjust confounding factors for every single cell and a random effect term for repeated sampling of a single donor. Indeed, Nathan et al. successfully deploy a PME model to a large-scale single-cell dataset comprising more than 500,000 unstimulated memory T cells from 259 donors [35]. This study demonstrated the utility of the PME model single-cell eQTL analysis to detect the cell-state dependency of eQTL effects. Using this model, they successfully showed that risk variants of autoimmunity were enriched in cell-state-dependent eQTLs (e.g., ORMDL3 and CTLA4 loci), indicating that cell-state context is crucial to understanding the genetic etiology of autoimmunity.

Although previous eQTL studies substantially contributed to a better understanding of autoimmunity pathology, these studies have primarily focused on the quantitative aspect of gene expression. However, its qualitative aspect is also critical for cellular biology and the immune system. RNA splicing is crucial to enhance the complexity of protein sequences and functions, and almost all genes have splicing isoforms. Therefore, splicing quantitative trait loci (sQTL) may illuminate autoimmunity pathology not explained by eQTL alone. sQTL analytical strategy is much more complicated than eQTL; we summarized sQTL methods used in the previous studies (Table 1).

One of the apparent challenges in splice isoform quantification is that most RNA-seq platforms are short-read sequencing. Typically, the read pair only covers a few hundred bases at most, whereas the median length of mRNAs is around 3000 base pairs [36]. Therefore, we cannot directly capture the entire isoform structure in most cases using short-read sequencing. On the other hand, we can directly capture splice junctions even using short-read sequencing.

**Table 1 genes-14-00572-t001:** Software for splicing isoform quantifications.

Software	Year	Method	Annotation	Novel Isoform Detection	Features
LeafCutter [37]	2018	Event	Not required	Yes	Focused on the variation in “intron” splicing. Used in many sQTL studies. Computationally efficient and accurate at detecting splicing events.
DEXSeq [38]	2012	Event	Required	No	Focused on differentially used exons.
rMATS [39]	2014	Event	Required	Yes	Analyzes replicate RNA-seq data. Accounts for sampling uncertainty and variability.
SUPPA2 [40]	2015	Event	Required	No	High accuracy at low sequencing depth and short read length.
MAJIQ [41]	2016	Event	Required	Yes	Designed to detect “complex” splice variations (e.g., alternative splice site and intron retention)
Cufflinks [42]	2012	Isoform	Not required	Yes	Early-phase software developed in 2010. A transcriptome assembler (it can estimate novel isoform structures). A successor software (stringTie) has already been developed.
StringTie2 [43]	2019	Isoform	Not required	Yes	Capable of assembling both short and long reads. Higher accuracy for assembling complicated isoforms (those with many exons) than Cufflinks.
RSEM [44]	2011	Isoform	Required	No	Available for organisms lacking sequenced genomes. Computationally intensive.
Salmon [45]	2017	Isoform	Required	No	Fast quantification due to alignment-free quantification. Accounts for sample-specific bias.
Kallisto [46]	2016	Isoform	Required	No	Fast quantification due to alignment-free quantification. Pseudoaligns the reads to the reference avoiding alignment of individual bases.

Year, the year of publication; method, the method of splice isoform quantification (either of splice event- or isoform-level quantification); annotation, requirements of annotation files (e.g., GTF file); novel isoform detection and the ability to detect novel splice isoform(s).

Leafcutter is a leading software widely used for splicing event detection [37]. Leafcutter extracts introns from reads that span between two exons from each sample integrates these across samples and defines a group of introns that share at least one splice site as an intron cluster. Leafcutter then calculates an intron excision ratio for each sample. Changes in this ratio provide a quantitative view of splicing changes. Leafcutter has been used in numerous studies, particularly in sQTL studies [47].

Leafcutter has multiple advantages over other splicing detection methods. Leafcutter does not require an existing annotation file, allowing for identifying novel splicing events. In addition, while other methods for quantifying exons (DEXSeq [38], rMATS [39], and SUPPA2 [40]) are unstable due to ambiguity in assigning reads that map to multiple isoforms of a gene, Leafcutter solves this problem by quantifying introns instead of exons. On the other hand, Leafcutter has several disadvantages. We cannot directly compare the Leafcutter results from different datasets because the definition of intron clusters is dataset-dependent. In addition, relating splicing events to transcript-level quantification is often tricky.

Instead of detecting splicing events at the exon junctions, we can computationally estimate the abundance of full-length transcripts from short-read sequence data (e.g., RSEM [44] and Cufflinks [42]), although the accuracy is relatively low. We can use these estimates to test the associations between the isoform usage ratio and genetic variants. For example, we used Cufflinks in our previous study and found an intriguing sQTL signal; rs10466829, a multiple sclerosis risk variant, showed an sQTL effect on *CLECL1* without noticeable eQTL effect in B cells [31]. This unique pattern (sQTL without eQTL) reflects that the expression of two major isoforms of *CLECL1* (NM_001267701 and NM_172004) were oppositely correlated with the risk variant. These isoforms differ only in the five amino acid residues at the extracellular domain of *CLECL1*. As exemplified by this result, sQTL studies can narrow candidate molecular etiology to specific molecule positions.

Inaccurate isoform quantification is partially caused by incomplete reference datasets we use for isoform quantification [48]. For example, some disease-causing isoforms have incomplete coding sequences in the GENCODE annotation [49]. Furthermore, even if all constituent exons are identified, complete isoform reconstruction from short-read data remains challenging [50].

In contrast to short-read sequencing, long-read sequencing techniques can generate reads of 10 kb or more and sequence full-length isoforms [51]. In one of our recent studies, we obtained a full isoform picture of the *PADI4* gene using long-read sequencing and found a novel non-functional splicing isoform lacking a functional domain [1]. With this updated *PADI4* isoform reference data, we re-analyzed one of our short-read sequencing datasets (Ref. [31]) and quantified *PADI4* isoform abundance. The splicing QTL signal for this novel *PADI4* isoform colocalized with the RA-GWAS signal [1]. This research direction is currently expanding. Inamo et al. performed long-read sequencing to create a complete isoform reference panel of fine-sorted immune cells, which improves the quality of future sQTL studies using immune cells (https://www.biorxiv.org/content/10.1101/2022.09.13.507708v1, accessed on 2 January 2023).

One of the most challenging and scientifically intriguing questions researchers have been asking is the cell type or tissue specificity of genetic effects. The GTEx v8 project shows that cis-sQTLs were significantly more tissue-specific than cis-eQTLs when considering all mapped cis-QTLs [52]. However, this pattern is reversed when considering only those cis-QTLs where the gene or splicing event is quantified in all tissues. This observation indicates that splicing measures are more tissue-specific than gene expression; in contrast, genetic regulation on splicing tends to be more shared, which suggests that it might be better to use the same cell types or tissues to investigate the effect of splicing on traits.

## 2. Discussion

As we introduced in this manuscript, V2F studies successfully identified several candidate causal mechanisms of the risk variants. However, many risk variants remain functionally characterized. Chun et al. evaluated how much of the autoimmunity risk variants can be explained by eQTLs discovered in the previous studies analyzing three major immune subpopulations [53]. To this end, they developed a new analytical method called joint likelihood mapping (JLIM) and found that eQTL signals only account for around 25% of the risk loci. Although sQTL can explain an additional fraction of heritability independent from eQTL, the gain in the ratio is relatively limited [54]. To further evaluate the eQTL-mediated autoimmunity genetic risk, Yao et al. developed a sophisticated method called mediated expression score regression (MESC) that accounts for genome-wide GWAS and eQTL signals [55]. They applied MESC to GWAS results for Crohn’s disease and eQTL results obtained in immune cells and found that gene expression levels mediated only around 20% of heritability. If we assume all non-coding risk variants possess eQTL or sQTL in specific immune cell types (although we admit this is an over-simplified scenario), these results suggest that the previous QTL projects have failed to detect such QTL signals; we call this “missing QTL”.

How can we solve the missing QTL problem? The straightforward approach will be diversifying the cellular conditions (e.g., various stimulatory conditions) where we test eQTL and sQTL. In addition, we can use single-cell transcriptomes to improve cellular resolution. However, of course, the culprit may be other molecular phenotypes, not expression and splicing, such as RNA editing [56] and other omics (e.g., metabolomics). Large-scale functional genomic experiments may not be a single solution. For example, the recent rapid progress of machine learning technologies started to solve the regulatory codes in our genome [57,58], i.e., we can partially infer the variant’s function solely based on the genomic sequence patterns around that variant. At the moment, we have not yet reached a conclusion about what the best approach is to maximize biological information extracted from GWAS outputs. In any case, we need to scale up V2F studies further.

## Figures and Tables

**Figure 1 genes-14-00572-f001:**
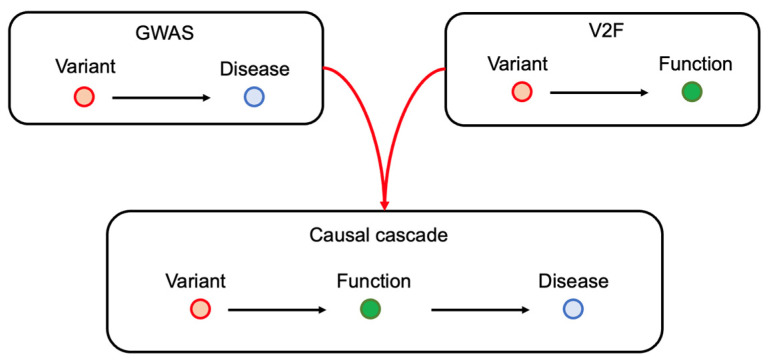
The V2F study illuminates the causal cascade of autoimmunity. The GWAS connects variants and diseases. V2F studies link variants and function. The function can be any immune-related phenotypes. The most studied and feasible phenotype is gene expression levels in immune cells. By combining GWAS and V2F outputs, we can draw the causal cascade.

**Figure 2 genes-14-00572-f002:**
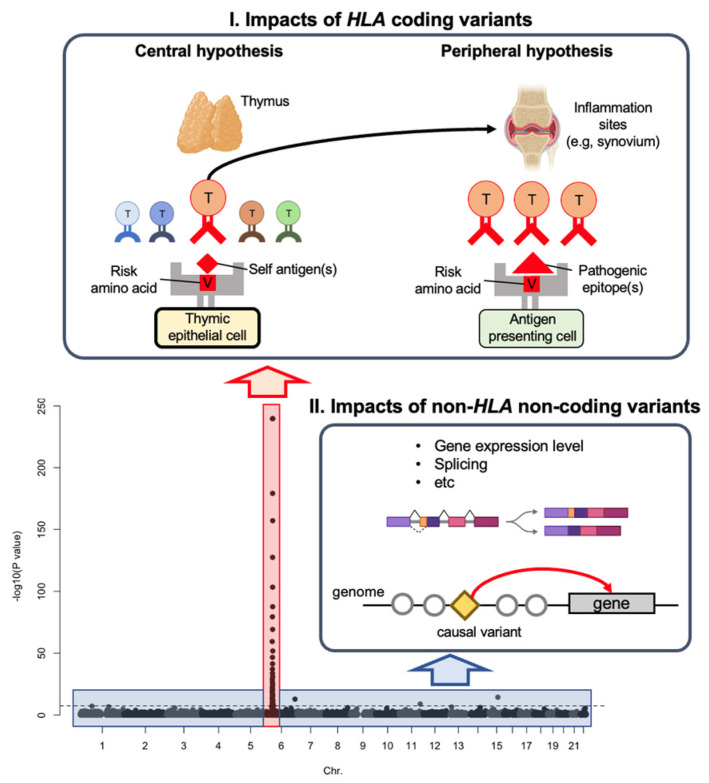
Two genetic risk categories of autoimmunity. In this review article, we introduced two categories of genetic risk of autoimmunity. One of our recent studies suggested that *HLA* coding variants influence thymic selection, modify TCR repertoire, and increase the auto-reactive immune response (the “central hypothesis”) [13]. Other researchers reported that *HLA* coding variants influence the binding affinity of pathogenic epitopes and enhance immune reactions against them (the “peripheral hypothesis”). On the other hand, non-*HLA* non-coding variants are enriched in the regulatory region and probably influence gene expression and splicing. We used *p*-values in our recent multi-ancestry of RA-GWAS for the bottom Manhattan plot [1]. We used images of the thymus, joint, and splice isoforms from BioRender (https://biorender.com/, accessed on 2 January 2023).

## Data Availability

Not applicable.

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
