# Peer review of "Functional Genetics to Understand the Etiology of Autoimmunity"

_genes, 2023, doi:10.3390/genes14030572_

Round 1
Reviewer 1 Report
This manuscript reviews the variant-to-function (V2F) studies on etiology of human autoimmune diseases. The text is a generally easy-to-read. We have three main suggestions and concerns -
1. How were the reviewed papers selected? The authors need to clearly describe their criteria. Ideally, they should provide a PRISMA description (https://www.prisma-statement.org).
2. Many recent reviews on this topic were not mentioned (see below). It is unclear what unique perspective is offered by the current review.
https://www.sciencedirect.com/science/article/abs/pii/S0896841122001305
https://www.sciencedirect.com/science/article/pii/S0002929721000938
https://ard.bmj.com/content/80/6/689.info
https://www.ncbi.nlm.nih.gov/pmc/articles/PMC7530520/
https://link.springer.com/article/10.1007/s00281-022-00915-x
3. Likewise, many classical reviews on this topic were not mentioned at all -
https://www.ncbi.nlm.nih.gov/pmc/articles/PMC4889885/
https://www.ncbi.nlm.nih.gov/pmc/articles/PMC4896831/
https://www.nature.com/articles/jhg201594
https://www.nejm.org/doi/full/10.1056/nejmra1100030
Author Response
Thanks for providing us with positive comments.
1. How were the reviewed papers selected? The authors need to clearly describe their criteria. Ideally, they should provide a PRISMA description (https://www.prisma-statement.org).
Thanks for this comment. We understand that "structured reviews and meta-analyses should use the same structure as research articles and ensure they conform to the PRISMA guidelines" according to the submission guideline for the authors. However, we did not intend to provide a structured review or meta-analysis but rather to provide our "expert opinions" and views on functional genetics. Therefore, we do not think we have to follow the rules for structured reviews.
2. Many recent reviews on this topic were not mentioned (see below). It is unclear what unique perspective is offered by the current review.
Thanks for this comment. We do not think we have to refer to other "review" articles in our review article.
3. Likewise, many classical reviews on this topic were not mentioned at all -
Thanks for this comment. We do not think we have to refer to other "review" articles in our review article.
Reviewer 2 Report
The authors describe that they recently developed a novel analytical pipeline of T cell receptor (TCR) repertoire to understand how HLA coding variants influence the genetic risk to autoimmunity.
I consider that the topic of the article is interesting. However, there are the following comments:
1) Include more studies that describe the findings of the studies conducted integrative analyzes using expression quantitative trait loci (eQTL) and splicing quantitative trait loci (sQTL), which could demonstrate that the risk non-coding variants dysregulate specific genes' expression and splicing, this is because manuscript leans towards the findings made by the authors.
2) Include a discussion section in the manuscript.
3) It is difficult for me to identify what type of review article it is: For example, a review of the literature or a critical review. Mention how the selection of articles for this review was made and what type of review it is. A table is needed to compile the methodology's search terms and databases.
4) Inclusion and exclusion criteria need to be stated, along with the number of articles identified.
5) Please refer to PRISMA guidelines for reporting literature review and updated methods accordingly.
6) Provide search strategy as supplementary material.
7) Provide a flow diagram for the inclusion/exclusion of final articles in this study.
Minor comments:
1) In paragraph 27 define V2F, in the same way at the bottom of figure 1, define GWAS and V2F.
2) Figure 2 should be more explanatory, the upper part of the figure should be marked as figure 1(a) and the lower part as 1(b) and each event should be explained at the bottom of the figure.
Author Response
Thanks for providing us with positive comments.
1) Include more studies that describe the findings of the studies conducted integrative analyzes using expression quantitative trait loci (eQTL) and splicing quantitative trait loci (sQTL), which could demonstrate that the risk non-coding variants dysregulate specific genes' expression and splicing, this is because manuscript leans towards the findings made by the authors.
Thanks for this comment. In the revised manuscript, we added additional examples of eQTLs and sQTL studies, putting an emphasis on single-cell eQTL.
2) Include a discussion section in the manuscript.
We added a discussion section.
3) It is difficult for me to identify what type of review article it is: For example, a review of the literature or a critical review. Mention how the selection of articles for this review was made and what type of review it is. A table is needed to compile the methodology's search terms and databases.
4) Inclusion and exclusion criteria need to be stated, along with the number of articles identified.
5) Please refer to PRISMA guidelines for reporting literature review and updated methods accordingly.
6) Provide search strategy as supplementary material.
7) Provide a flow diagram for the inclusion/exclusion of final articles in this study.
Thanks for this comment. Regarding comments 3-7, we understand that "structured reviews and meta-analyses should use the same structure as research articles and ensure they conform to the PRISMA guidelines" according to the submission guideline for the authors. However, we did not intend to provide a "structured review" or meta-analysis but rather to provide our "expert opinions" and views on functional genetics. Therefore, we do not think we have to follow the rules for structured reviews.
Minor comments:
1) In paragraph 27 define V2F, in the same way at the bottom of figure 1, define GWAS and V2F.
2) Figure 2 should be more explanatory, the upper part of the figure should be marked as figure 1(a) and the lower part as 1(b) and each event should be explained at the bottom of the figure.
For minor comments 1-2, we noticed it is due to the bug in the file formatting in the editorial office. The original figure legends were at the bottom of the main text. In the revised manuscript, they are moved to the appropriate places.
Round 2
Reviewer 2 Report
The authors have addressed the comments made to improve the paper.
minor comment:
It only remains to define the acronym for V2F between parentheses in the document's text where it is mentioned for the first time.